# Modeling Aversion Resistant Alcohol Intake in Indiana Alcohol-Preferring (P) Rats

**DOI:** 10.3390/brainsci12081042

**Published:** 2022-08-05

**Authors:** Simon N. Katner, Alena M. Sentir, Kevin B. Steagall, Zheng-Ming Ding, Leah Wetherill, Frederic W. Hopf, Eric A. Engleman

**Affiliations:** 1Department of Psychiatry, Institute of Psychiatric Research, Indiana University School of Medicine, Indianapolis, IN 46202, USA; 2Stark Neuroscience Research Institute, Indiana University School of Medicine, Indianapolis, IN 46202, USA; 3Departments of Anesthesiology and Perioperative Medicine and Pharmacology, Pennsylvania State University College of Medicine, 700 HMC Crescent Road, Hershey, PA 17033, USA; 4Department of Medical and Molecular Genetics, Indiana University School of Medicine, Indianapolis, IN 46202, USA

**Keywords:** addiction, compulsive drinking, genetic model, alcoholism, selected lines, alcohol preference, gender studies

## Abstract

With the substantial social and medical burden of addiction, there is considerable interest in understanding risk factors that increase the development of addiction. A key feature of alcohol use disorder (AUD) is compulsive alcohol (EtOH) drinking, where EtOH drinking becomes “inflexible” after chronic intake, and animals, such as humans with AUD, continue drinking despite aversive consequences. Further, since there is a heritable component to AUD risk, some work has focused on genetically-selected, EtOH-preferring rodents, which could help uncover critical mechanisms driving pathological intake. In this regard, aversion-resistant drinking (ARD) takes >1 month to develop in outbred Wistar rats (and perhaps Sardinian-P EtOH-preferring rats). However, ARD has received limited study in Indiana P-rats, which were selected for high EtOH preference and exhibit factors that could parallel human AUD (including front-loading and impulsivity). Here, we show that P-rats rapidly developed compulsion-like responses for EtOH; 0.4 g/L quinine in EtOH significantly reduced female and male intake on the first day of exposure but had no effect after one week of EtOH drinking (15% EtOH, 24 h free-choice paradigm). Further, after 4–5 weeks of EtOH drinking, males but not females showed resistance to even higher quinine (0.5 g/L). Thus, P-rats rapidly developed ARD for EtOH, but only males developed even stronger ARD with further intake. Finally, rats strongly reduced intake of quinine-adulterated water after 1 or 5 weeks of EtOH drinking, suggesting no changes in basic quinine sensitivity. Thus, modeling ARD in P-rats may provide insight into mechanisms underlying genetic predispositions for compulsive drinking and lead to new treatments for AUDs.

## 1. Introduction

Alcohol use disorder (AUD) is a pervasive [1,2] and costly health issue, with one in four Americans having an AUD in their lifetime and exacting a financial burden of nearly a quarter of a trillion dollars per year in the U.S. [3,4,5,6,7,8,9]. Additionally concerning, is that problem alcohol (EtOH) drinking in women has risen dramatically in recent years [1,10,11], and women can have greater alcohol problems [12,13,14]. Moreover, prenatal EtOH exposure is known to have effects on embryonic development and related neurological deficits, which, along with cross fetal-parental genetic and epigenetic factors may contribute to compulsive EtOH use [15,16,17,18]. Furthermore, decades of research provide strong evidence that heritable factors can play a pivotal role in the development of AUD, as individuals positive for a family history of alcoholism have a significantly greater chance of developing AUDs [19,20,21,22,23,24]. However, despite the importance of heritable factors for AUD, much has remained unclear about the critical underlying differences that promote AUD development. In addition, there are very few effective treatments currently available to treat AUDs despite the clear and urgent need [25,26,27], with relapse rates ranging from 50 to 80% [8,28,29].

Compulsion-like alcohol drinking, where intake persists despite negative consequences, can be a particular obstacle to treatment and a strong driver of excessive intake [30,31,32,33,34,35,36,37,38,39], in part accompanied by a decrease in volitional control [40,41,42]. Intake, despite negative consequences, features prominently in the DSM-V definition of AUD (e.g., [43,44]), and several groups find less sensitivity to cost with AUD [45,46,47] (while one group does not [48,49], see Discussion), while greater drinking levels are associated with more alcohol problems [44,50,51,52].

Compulsion-like, aversion-resistant drinking (ARD) has been modeled in rodents by the persistence of intake despite EtOH being paired with quinine or footshock [36,38]. Quinine- and footshock-resistant alcohol consumption are mediated by a similar cortico-accumbens circuit [53], and rats with higher shock resistance also have greater quinine resistance [54,55], together validating the use of the technically simpler quinine (which is also easier to test in a graded manner vs. shock). In addition, rodents tolerate quinine levels in alcohol that are strongly avoided in water [56,57], confirming aversion-resistance for alcohol [38]. Willingness to drink quinine-adulterated EtOH is also associated with the transition to alcohol dependence in rats [58,59] and mice [60]. Finally, several studies using outbred Wistar rats show that the development of quinine resistance takes several months of intermittent access intake [53,56,61] (see Discussion).

However, the vast majority of rodent ARD studies use outbred rats and thus do not model the heritable factors associated with a family history of alcoholism, which, as noted above, significantly increases the risk of transitioning to AUD [62]. The Indiana University rodent lines selected for alcohol preference (and in particular the P-line) are the best characterized and documented genetic animal models of alcoholism currently available [63,64,65]. P-rats have proven quite useful in providing a better understanding of heritable neurobiological and behavioral phenotypes that can contribute to excessive EtOH drinking. For example, P-rats show greater impulsivity than their Wistar progenitor [66], and impulsivity is a risk factor for excessive intake in humans (e.g., [67,68,69]). P-rats also have strong front-loading [70], where a high initial intake of alcohol can indicate high motivation [71,72,73,74]; faster alcohol exposure in humans relates to the risk of developing AUD [75] and high-risk drinking [76], and rapid responding to alcohol is associated with genetic risk [77]. Thus, it would be particularly interesting if P-rats also exhibit a propensity for high ARD, especially accelerated development of ARD for EtOH, which we pursue here.

## 2. Materials and Methods

### 2.1. Subjects and Conditions

A total of 87 adult female (*n* = 39; weighing approximately 250 g at the beginning of the study) and male (*n* = 48; weighing approximately 350 g at the beginning of the study) alcohol-preferring (P) rats were used in this study. Rats were individually housed beginning ~PND63 in plastic cages with wood-chip bedding, with free access to rat chow and water, in a reverse 12 h light/dark cycle (lights off at 08:00 h). Intake studies began at least 3 days after adaptation to single housing. Animal weights were measured twice weekly and the average weights were used to determine intake as g EtOH per kg body weight. All experiments were conducted with the approval of the Indiana University School of Medicine Animal Care and Use Committee (IACUC) and in compliance with the Guide for Laboratory Animal Care and Use.

### 2.2. Alcohol and Quinine

EtOH was diluted from 95% EtOH stock solutions in deionized water. Quinine hydrochloride was dissolved in either deionized water or 15% (*v*/*v*) EtOH and expressed as grams per liter (g/L). Unless otherwise stated, rats received 24 h access to EtOH (+/− quinine) in a free-choice drinking paradigm, with food and water always available; all EtOH drinking in this study utilized 15% EtOH.

### 2.3. Quinine in EtOH Sensitivity Test

EtOH-naïve female P-rats (*n* = 6) were given access to EtOH mixed with increasing concentrations of quinine (0.1, 0.2, 0.3, and 0.4 g/L) on consecutive days. Each quinine-EtOH concentration was presented for 24 h, and intake was measured each day. A control group of female P-rats (*n* = 5) had access to 15% EtOH only over the same period, with EtOH intake averaged across the days of intake.

### 2.4. Effect of 0.4 g/L Quinine on the First Day of EtOH Drinking

EtOH-naïve male (*n* = 5) and female (*n* = 8) P-rats were presented with EtOH mixed with 0.4 g/L quinine for 24 h. Intake was compared with controls drinking EtOH without quinine for one day (male *n* = 5; female *n* = 7). In addition, quinine-EtOH rats drank EtOH-only the following day to assure that these rats would drink normal EtOH levels.

### 2.5. Effect of 0.4 g/L Quinine after One Week EtOH Drinking

To determine if one week of EtOH intake would produce resistance to quinine, male (*n* = 11) and female (*n* = 12) P-rats were given 1-week of continuous access to EtOH. The following day, rats were allowed to drink 0.4 g/L quinine in EtOH.

### 2.6. Effects of Chronic EtOH Drinking on Resistance to 0.5 g/L Quinine in EtOH

Anecdotal evidence from lab investigators suggested that P-rats with several weeks of EtOH drinking history may develop aversion resistance to higher concentrations of quinine. Therefore, for this experiment, male (*n* = 12) or female (*n* = 12) P-rats were given 3 weeks of free-choice access to EtOH. At the end of the third week, EtOH-only was replaced with EtOH plus 0.5 g/L quinine in one 24 h period. Rats then had free-choice drinking of EtOH for a fourth and fifth week, with a retest of sensitivity to 0.5 g/L quinine in EtOH during the final 24 h at the end of each week.

### 2.7. Effect of 1 Week of Scheduled Access EtOH Drinking, after 1 or 5 Weeks of Continuous-Access EtOH Drinking, on Sensitivity to 0.5 g/L Quinine in EtOH

To determine if a one-week period of scheduled access drinking (1 h of access per day) would impact the development of aversion resistance to 0.5 g/L quinine, male P-rats from experiment 2.6 that drank EtOH under continuous access for 5 weeks were compared with a group with continuous access for only one week. Afterward, rats had 5 days of limited access to drinking (1 h/day), and, thereafter, were tested for sensitivity to 0.5 g/L quinine in EtOH using the 24 h free-choice protocol.

### 2.8. Sensitivity to Quinine in Water Test

Male (*n* = 7, 8) P-rats were given continuous free-choice access to EtOH for either 1 week or 5 weeks. At the end of the EtOH access periods, rats were given the choice of two water bottles, one containing water only, the other with 0.1 g/L of quinine dissolved in water, and intake was measured over 24 h. Intake from the two bottles was compared to determine if the rats had become insensitive to, or possibly even preferred, the taste of quinine.

### 2.9. Statistics

Data are presented as the mean ± SEM. Data were analyzed using Student’s t-test or analyses of variance (ANOVAs) with or without repeated measures (RM), or mixed-effects analysis, when indicated, followed by post-hoc tests where appropriate. Graphs and analyses were generated using GraphPad Prism 8 (GraphPad Software, La Jolla, CA, USA).

## 3. Results

### 3.1. Sensitivity to Quinine in EtOH Early in Drinking History

We first tested the hypothesis that P-rats might develop compulsion-like responses after a brief history of EtOH intake. Thus, female P-rats were given 15% EtOH mixed with increasing concentrations of quinine over four successive days. Our preliminary experiments indicated that P-rats without EtOH drinking experience did not show sensitivity to 0.1 g/kg quinine in EtOH in a 24 h 2 bottle choice paradigm. Therefore, quinine concentrations in EtOH began at 0.1 g/L and increased by 0.1 g/L each day, until a significant reduction in EtOH drinking was observed. A control group had free choice access to 15% EtOH over the same period (Figure 1). Under this paradigm, increasing quinine concentration in 15% EtOH to 0.4 g/L quinine significantly reduced 24 h intake of 15% EtOH [* *p* < 0.01, RM ANOVA, F (3, 23) = 8.73, *n* = 6]. A Tukey’s posthoc multiple comparisons test found that level of 0.4 g/L of quinine significantly reduced intake of 15% EtOH compared with intakes with 0.1 [*p* < 0.01], 0.2 [*p* < 0.05], or 0.3 g/L [*p* < 0.05] quinine. For comparison, the mean 4-day intake of EtOH in a separate set of control rats is presented (*n* = 5). No differences in weight gain over the testing period were observed between animals in the quinine testing group (+4.0 ± 1.3 g) and the control group (+4.8 ± 2.5 g), respectively (*t* = 0.82; *t*-test; *p* > 0.05).

### 3.2. Quinine Effects on the First Day of EtOH Drinking

We next examined whether 0.4 g/L quinine in EtOH would reduce intake on the first day of EtOH exposure. Indeed, this quinine dose significantly reduced EtOH drinking in both female and male rats (relative to separate control P-rats drinking only 15% EtOH; Figure 2). Specifically, adulteration with 0.4 g/L quinine robustly reduced the intake of EtOH in males (*n* = 5; *t* = 11.09, *p* < 0.001, paired t-test vs. white bars; *t* = 6.63, *p* < 0.001 vs. control group; unpaired *t*-test) and females (*n* = 7; *t* = 2.84, *p* < 0.03, paired *t*-test vs. white bars; *t* = 2.94; *p* < 0.03 vs. control group; unpaired *t*-test). Moreover, the following day, when the EtOH was presented without quinine, rats drank significantly more EtOH without quinine relative to EtOH-quinine, and at levels not different from controls (Figure 2).

### 3.3. Effects of 1 Week of EtOH Drinking on Resistance to Quinine in EtOH

We next examined whether one week of continuous EtOH drinking was sufficient to induce quinine-resistant EtOH consumption. Indeed, after one week of intake, both female and male P-rats were fully resistant to 0.4 g/L quinine in EtOH, such that the EtOH-quinine intake did not differ from the mean weekly intake of EtOH only (Figure 3A males, *n* = 11, *t* = 0.88, *p* > 0.05; Figure 3B females *n* = 12, *t* = 0.74, *p* > 0.05; females). This suggests that P-rats developed quinine resistance quicker than other rat strains tested in the literature and are perhaps more similar to C57BL/6 mice with their rapid development of aversion resistance (see Discussion).

### 3.4. Effects of Chronic EtOH Drinking on Resistance to 0.5 g/L Quinine in EtOH

Since one-week EtOH intake history led to resistance to 0.4 g/L quinine in EtOH, we next examined whether additional weeks of EtOH drinking (total 5 weeks of intake) would lead to even greater quinine resistance, in particular by testing intake of EtOH plus 0.5 g/L quinine at the end of each week. We found that male P-rats were sensitive to this very high quinine level in EtOH after 3 weeks of drinking, but males became resistant to this high concentration of quinine after 4 or 5 weeks of intake (Figure 4A). A three-way RM ANOVA (sex X EtOH ± quinine X week) revealed significant main effects of sex [F (1, 22) = 38.98; *p* < 0.001] and EtOH ± quinine [F (1, 22) = 15.94, *p* < 0.001] and a week X EtOH ± quinine interaction [F (2, 44) = 12.12, *p* < 0.001]. A Tukey’s multiple comparisons posthoc test revealed that for males, the 0.5 g/L quinine significantly reduced EtOH intake in week 3 (* *p* < 0.05), but not in weeks 4 and 5 (*p* > 0.05). In strong contrast, female rats showed a significant reduction of EtOH intake when adulterated with 0.5 g/L quinine in the third (*p* < 0.001), fourth (*p* < 0.01), and fifth (*p* < 0.001) week of EtOH drinking (Figure 4B). Together, our results thus far suggest that P-rats rapidly developed a significant level of aversion-resistant EtOH intake, but that only males developed even stronger compulsion-like responses with further weeks of EtOH drinking.

### 3.5. Effect of Scheduled Access EtOH Drinking on Quinine Resistance

Some studies have suggested that limited daily access or intermittent access can facilitate the development of pathological drive for EtOH [56,78]. In addition, the continuous drinking method used here is not easily amenable to pharmacological or other experimental interventions. Moreover, it is unclear if a change in the drinking paradigm and/or additional EtOH exposure might affect the status of aversion resistance in animals displaying either quinine resistance or sensitivity as observed in the above experiment. Thus, we examined whether aversion-resistant patterns were also observed in a hybrid model where rats had 1 or 5 weeks of continuous EtOH intake, then five days of 1 h /day access to EtOH (tested only in male rats, since they showed high quinine-resistance after 5 weeks intake). Both groups acquired EtOH drinking in the first 1-h session and averaged 1.1 ± 0.1 g/kg/1 h (*n* = 8, one-week EtOH history) and 1.0 ± 0.1 g/kg/1 h (*n* = 12, five weeks EtOH history) (no between-group differences).

Interestingly, after one week of continuous drinking plus 5 days of 1 h/day access, EtOH drinking was strongly reduced by including 0.5 g/L quinine in the EtOH (Figure 5A), similar to the sensitivity observed after one week of drinking without limited daily access (Figure 4A). However, like responding after 5 weeks of continuous drinking but no limited access days (Figure 4B), intake after 5 weeks of continuous access plus 5 days of limited access was strongly resistant to adulteration with 0.5 g/L (Figure 5B). Thus, our drinking model can include brief training in 1 h/day sessions while retaining the time course of development of resistance to very high quinine, setting a strong foundation for future studies of pharmacological and other mechanistic interventions to counteract compulsion-like drives for alcohol.

### 3.6. Quinine in Water Sensitivity Test

One possibility that could explain the development of aversion resistance, i.e., tolerating aversive quinine in the EtOH, is that weeks of EtOH drinking leads to more basic changes in quinine sensitivity. Thus, we tested whether sensitivity to 0.1 g/L quinine in water changed after one or five weeks of EtOH drinking (*n* = 7, 8/group). We chose this lower dose because P-rats will drink 0.1 g/L quinine in EtOH in the very first drinking session (Figure 1), perhaps suggesting that P-rats are innately more aversion resistant in terms of EtOH drinking behavior. Additionally, testing with 0.4 g/L in water might lead to such strong reductions in intake that more subtle changes in quinine sensitivity might not be apparent. In particular, we found that 0.1 g/L quinine in water strongly reduced intake, with a similar effect after one or five weeks of EtOH drinking (Figure 6). These findings are important because they suggest that (1) P-rats can sense and respond to lower quinine doses; (2) 0.1 g/L is, in itself, highly aversive to P-rats (and that higher quinine doses are also likely to be strongly aversive) and (3) tolerance of high quinine levels in EtOH suggests the development of compulsion-like promotion of EtOH intake. No differences in body weight gain were observed in animals receiving EtOH for one week vs. five weeks.

## 4. Discussion

The current study compared the development of ARD using the 24 h free-choice intake of quinine-adulterated 15% EtOH in female and male P-rats. The introduction of increasing concentrations of quinine did not reduce EtOH intake in females until 0.4 g/L quinine was reached. In addition, both females and males had significantly lower EtOH plus 0.4 g/L quinine drinking on the first day of access to EtOH, relative to intake of EtOH alone. However, after only one week of alcohol drinking, both male and female P-rats showed resistance to 0.4 g/L quinine in EtOH, maintaining higher intake levels. Thus, female and male P-rats were both sensitive to 0.4 g/L quinine reduction in EtOH drinking on the first day of access, with resistance to 0.4 g/L after one week of EtOH drinking. We then examined whether additional weeks of EtOH consumption would lead to greater ARD. Indeed, male rats showed resistance to 0.5 g/L quinine in EtOH, and maintained intake levels, after 4 and 5 weeks of EtOH drinking history, but not after 3 weeks. In striking contrast, females did not develop resistance to 0.5 g/L quinine in alcohol in the same paradigm. Finally, 0.1 g/L quinine significantly and strongly reduced intake of water after 1 or 5 weeks of EtOH consumption, suggesting that P-rats could sense and find quinine aversive, and also that a longer history of EtOH drinking did not produce large shifts in basal quinine sensitivity. Together with previous work where female and male rodents were equally sensitive to quinine in water (with or without EtOH drinking history) [57,79,80] and to shock [81], these findings suggest that heritable selection for EtOH preference in P-rats is associated with increased susceptibility to develop aversion resistance for EtOH. This point is particularly salient since the P-rat line was selected for alcohol preference over many generations and prenatal EtOH exposure is known to have effects on embryonic development and related neurological deficits, which, along with cross fetal-parental genetic and epigenetic factors may contribute to compulsive EtOH use [15,16,17,18].

As noted in the Introduction, the persistent drive to consume EtOH in the face of aversive consequences is a major obstacle in people with AUDs [30,31,32,33,34,35,36,37,38,39], and drinking EtOH in the face of negative, adverse consequences is characteristic of individuals with EtOH dependence [82], highlighting the importance of understanding mechanisms which promote compulsive-like, aversion resistant EtOH intake. Rodent models, in which animals voluntarily consume EtOH under aversive consequences, such as the addition of a bitter tastant (or footshock), enable researchers to study aversion-resistant EtOH drinking which models some aspects of AUD in humans, and such rodent models are particularly critical for uncovering the mechanisms driving this maladaptive behavior. Our studies suggest that P-rats with a brief history of EtOH intake demonstrate resistance to quite high levels of quinine in EtOH, while a lower level of quinine in water (0.1 g/L) was strongly avoided. In this vein, while P-rats and Wistars show similar levels of taste sensitivity and reactivity to quinine alone [83,84,85], EtOH intake in Wistars is greatly reduced by 0.1 g/L quinine in EtOH, even after >3 months of intermittent access EtOH intake, although Wistars exhibit ARD with 30 mg/L [56,71,72] and 60 mg/L quinine (unpublished) in EtOH. Similarly, 0.1 g/L quinine reduces EtOH intake in Wistar but not P-rats in a Pavlovian two-way cued access protocol paradigm [86]. However, in a different paradigm using an operant discriminative stimulus task, adulteration with 0.15 g/L reduces EtOH intake in both Wistar and P-rats [87], suggesting that ARD differences between rat lines can be affected by the behavioral paradigm (and see below).

Nonetheless, these data together suggest that selection for alcohol preference may impart an innate level of aversion resistance in addition to more rapid development of resistance to higher aversion. This highlights the potential for heritable contributions to the initial ARD level and the rate of further development of ARD with continued EtOH exposure. The prospect that compulsion-like drinking may in part reflect innate and possible heritable factors is supported by the identification of genes associated with such phenotypes in humans. For instance, the Met66BDNF polymorphism in humans is associated with elevated risk for certain substance use disorders [88,89,90], and transgenic expression of the mouse ortholog of this gene increases EtOH intake and aversion to resistant EtOH drinking [91]. These phenotypes are similar to those observed in P-rats, and further support the idea that genetics and family history may predispose individuals to compulsive substance use.

Of particular interest in our study is that P-rats developed ARD with only one week of EtOH drinking. In contrast, Wistar rats require 3 or more months of intermittent access to develop ARD [53,56,61], similar to Lister Hooded rats [92], and even Marchigian Sardinian P-rats show quinine-resistance after >1 month intermittent but not continuous alcohol intake [78]. Thus, Indiana P-rats may be particularly prone to the rapid development of ARD, at least using quinine. In this way, P-rats may be more similar to C57BL/6 mice, where several studies find that male mice can develop ARD after a few days to weeks of EtOH intake [93], or a few weeks [94,95]. Since both P-rats and C57BL/6 mice exhibit relatively high EtOH intake during their first drinking session, this suggests that higher levels of initial EtOH preference and voluntary intake may be associated with a more rapid expression of ARD.

Importantly, quinine-resistant drinking was not due to a lack of sensitivity to quinine as a tastant after high levels of EtOH exposure, as P-rats avoided quinine in water at concentrations several fold lower than the concentrations accepted in EtOH solutions, which was seen after one or five weeks EtOH drinking history (Figure 6). Moreover, previous data comparing quinine sensitivity in P vs. Wistar rats found no differences in taste reactivity [83,84], suggesting that the differences in the ability of quinine to reduce EtOH drinking between the lines are not due to a difference in quinine as a tastant, but rather the impact of the presence of EtOH. Interestingly, a study examining aversion-resistant alcohol seeking in P-rats using foot shock found that P-rats with extensive alcohol drinking experience diverged into groups (compulsive, intermediate, and non-compulsive) [35]. This study also found that the EtOH intake level was “not sufficient to predict the transition to compulsive alcohol-seeking behavior” [35]. Here, P-rats with the same amount of EtOH exposure did not divert into different aversion-resistant groups, suggesting that aversion-resistance in alcohol-seeking behavior in Giuliano and colleagues (2018) may identify individual differences not observed with aversion-resistant consumption behavior [79].

One interesting observation was that female and male P-rats both quickly developed resistance to 0.4 g/L quinine in EtOH, although only males developed further quinine resistance to 0.5 g/L quinine with additional weeks of EtOH-drinking history. As recently reviewed [80], female vs. male ARD has received some attention in recent times, with interesting, yet somewhat mixed findings. For example, several groups have found that female and male C57 mice had similar sensitivity to quinine in EtOH when consumed under bottle drinking [57,95,96]. In contrast, under continuous access, mouse EtOH intake is overall more sensitive to quinine than in other studies with intermittent access models, although females tolerate more quinine in EtOH than males [97]. Additionally, female mice show greater quinine resistance when tested under operant conditions [79], although female and male rats show similar quinine reduction in breakpoint for alcohol under progressive ratio [98]. In another study, female mice show little reduction in alcohol-conditioned place preference (CPP) when the alcohol-paired chamber is paired with footshock, while shock-pairing does reduce male CPP [81]. Interestingly, after exposure to EtOH vapor, females show greater sensitivity to shock relative to males in this model. Thus, taken together with our findings that males but not females show resistance to 0.5 g/L quinine after additional weeks of drinking, one possibility is that female ARD is more bimodal, where aversion resistance is similar to or greater than males with more limited EtOH history or lower levels of aversion, but females show less ARD than males with more extended EtOH experience and/or higher aversion, perhaps a type of protective factor. These possibilities remain highly speculative but are an important area for future studies to disentangle. We also note that the nature of mechanisms underlying sex differences in the development of higher ARD in P-rats remains unknown. While there are limited studies comparing female and male P-rat signaling in ARD, studies find no sex differences in several measures including naltrexone or cholinergic regulation of alcohol intake, or locomotor stimulation by low dose alcohol [99,100,101]. However, female P-rats can drink more alcohol under some conditions [102,103] but not others (including the present studies).

While compulsion-like, consequence-resistant responses to alcohol have the potential to contribute strongly to the expression of AUD, there are some important considerations to address. One is that negative consequences for humans are often in the future, unlike acute in rodent models. However, an alternate perspective is that, for treatment seekers, the negative consequences are more acute [30] (“actively struggling with the balance between known bad outcomes and the desire to drink” [80]), and we have always considered our findings to be more relevant for treatment seekers (since non-treatment seekers largely do not come for treatment). Thus, some clinicians have expressed support for the value of rodent compulsion-like alcohol drinking (CLAD) models [34,54,55,104,105]. In addition, while intake despite negative consequences is a prominent feature in the DSM-V AUD definition (e.g., [43,44]), and several groups find that people with AUD can have less sensitivity to cost [45,46,47], one recent group found that people with AUD are not more insensitive to cost for alcohol [48,49] which could question the importance of consequence-resistance. However, Epstein and Kowalczyk [34,105] have emphasized, from a clinician’s perspective, that multiple factors can contribute to AUD expression, with compulsion-like drives being one important factor.

In addition, while we have not directly addressed underlying mechanisms for compulsion-like alcohol responding, we note that the anterior insula cortex (AIC) is related to many aspects of alcohol drinking (reviewed in [106]). Some of these include where alcohol cue- [107,108] and negative affect-induced [109] AIC activation predict real-world drinking [110], AIC activation to alcohol images predicts later transition to heavy drinking [111], and therapies that reduce drinking also decrease AIC activity and/or connectivity (e.g., [112,113]). Importantly for the current work, rodent studies have shown that specific AIC-related circuits (including AIC projections to NAcb) and dmPFC are important for the expression of compulsion-like drinking (with little role in alcohol-only intake) [53,114,115,116]. Quinine-resistance is also associated with dysregulation in dmPFC glutamate systems [38,53,59]. In agreement, compulsion-like responding to alcohol in humans with AUD involves the activation of a similar aINS/mPFC/striatal circuit in both women and men [104,117]. Although not evaluated in this study, some evidence suggests that the estrous cycle may play a role in alcohol reinforcement [118], suggesting that aspects of AUD may be more complicated in females. Thus, while there is still much to be discovered about critical underlying mechanisms, the evidence thus far implicates the importance of compulsion-like drives in human AUD, with similar underlying ARD circuitry in humans and rodents, supporting the translational relevance of rodent models of compulsion-like responding.

## 5. Conclusions

The present findings show that both male and female P-rats, selectively bred for high voluntary EtOH intake, are initially sensitive to 0.4 g/L quinine in EtOH, while ARD to 0.4 g/L quinine develops after only one week of 24 h access to EtOH. In contrast, males became resistant to 0.5 g/L quinine by 4 or 5 weeks of EtOH drinking, while females did not. Together, our findings demonstrate that P-rats rapidly developed resistance to quinine in EtOH, yet retained aversion to quinine in water, suggesting that selection for EtOH preference is also associated with increased ARD level and faster ARD development. These findings provide a foundation for future research to employ the P-rat model to study the neurobiological basis of ARD and to provide insight into the mechanisms that underlie compulsive drinking.

## Figures and Tables

**Figure 1 brainsci-12-01042-f001:**
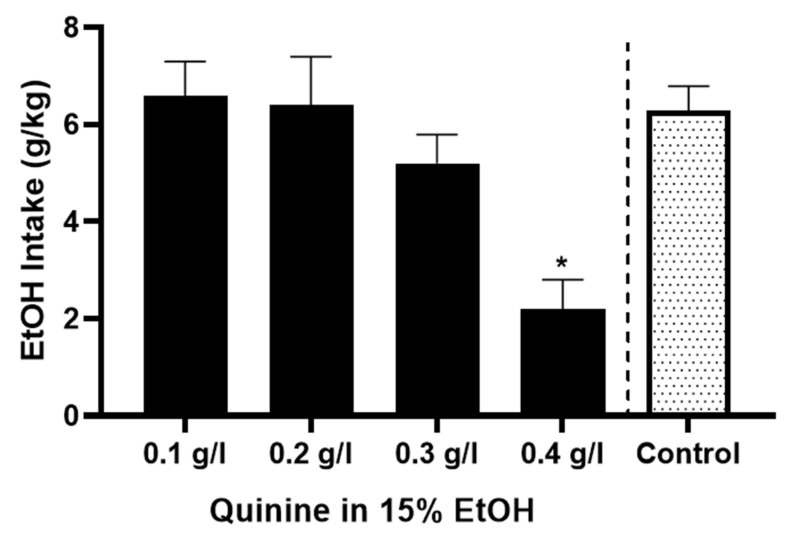
In alcohol naïve female P-rats (*n* = 6), using a daily escalation paradigm of increasing quinine concentration in 15% EtOH, 0.4 g/L quinine significantly reduced 24 h intake of 15% EtOH compared to intake of 0.1, 0.2, or 0.3 g/L quinine (* *p* < 0.01; *p* < 0.05 and *p* < 0.05, respectively, Tukey’s posthoc test). For comparison, the mean 4-day intake of EtOH in a separate set of control female P-rats drinking 15% quinine without EtOH is presented (*n* = 5).

**Figure 2 brainsci-12-01042-f002:**
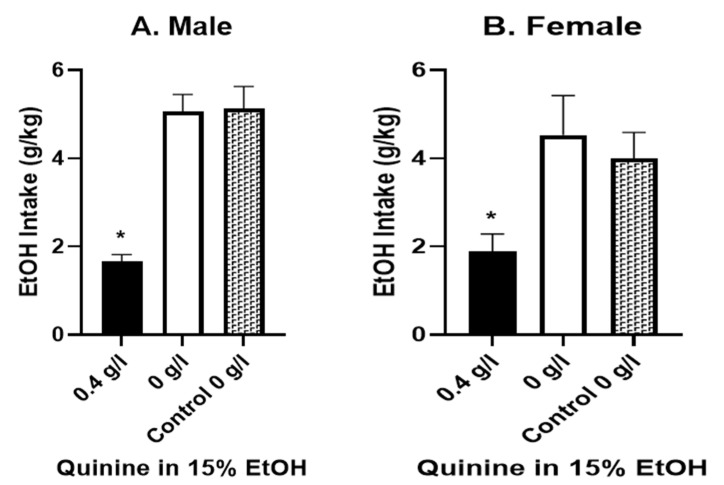
Quinine Effect on First Day of EtOH Drinking. Black bars: intake on the first day of exposure to 15% EtOH containing 0.4 g/L quinine; white bars: intake of 15% EtOH without quinine on the following day; hashed bars: intake of control rats (15% EtOH only) on the first day of EtOH access. Adulteration with 0.4 g/L quinine robustly reduced intake of EtOH in males (**A**) (*n* = 5; *t* = 11.09, * *p* < 0.001, paired *t*-test vs. white bar; * *p* < 0.001 vs. control group—hatched bar; unpaired *t*-test) and females (**B**) (*n* = 7; * *p* < 0.03, paired *t*-test vs. white bar; * *p* < 0.03 vs. control group—hatched; unpaired *t*-test).

**Figure 3 brainsci-12-01042-f003:**
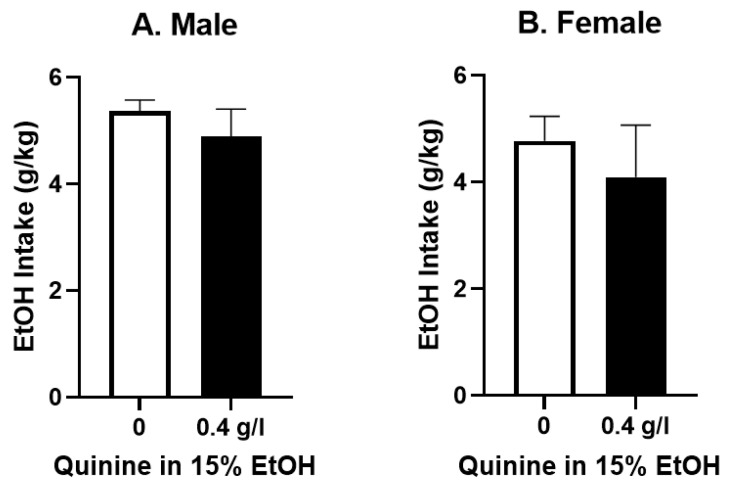
Quinine resistance after 1 week EtOH drinking: Male and female P-rats had one week 24 h continuous access to 15% EtOH. Average EtOH intake for that week is depicted in the open bar (zero quinine condition). The following day, 0.4 g/L quinine was added to EtOH (black bars), with no significant effect of 0.4 g/L quinine on EtOH intake in either male (**A**) or female (**B**) P-rats (*n* = 11, 12/group, *p* > 0.05, paired *t*-tests).

**Figure 4 brainsci-12-01042-f004:**
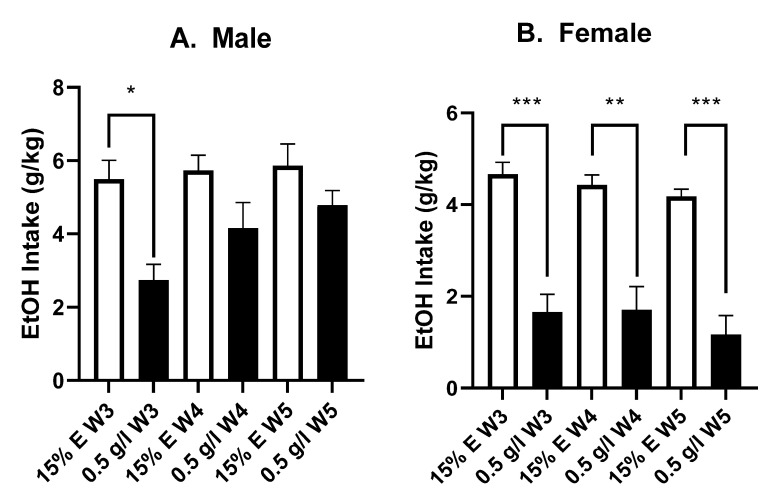
Quinine resistance to higher quinine (0.5 g/L) after 3–5 weeks EtOH drinking. (**A**) In male P-rats (*n* = 12)**,** EtOH drinking was sensitive to 0.5 g/L quinine after 3 weeks intake (W3) (* *p* < 0.05). However, after 4 weeks (W4) or 5 weeks (W5) of EtOH drinking, male intake persisted despite 0.5 g/L quinine in EtOH. (**B**) In contrast to males, female (*n* = 12) EtOH drinking was significantly reduced by 0.5 g/L quinine at W3 (*** *p* < 0.001), W4 (** *p* < 0.01), and W5 (*** *p* < 0.001).

**Figure 5 brainsci-12-01042-f005:**
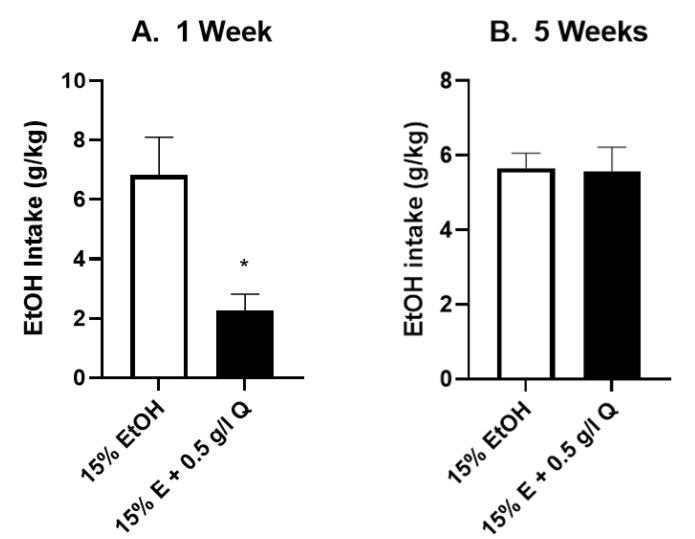
Adding one week of Limited Access Drinking does not alter Quinine Sensitivity. After (**A**) one week (*n* = 8) or (**B**) 5 weeks (*n* = 12) of continuous EtOH drinking, male P-rats had 5 days of limited access drinking (1 h/day), then were tested for sensitivity to 0.5 g/L quinine in EtOH using the 24 h free-choice protocol. (**A**) Rats with 1 week EtOH drinking history prior to daily limited access retained sensitivity to 0.5 g/L quinine in EtOH (* *p* < 0.05; paired *t*-test; *n* = 8). (**B**) However, rats with 5 weeks drinking history were resistant to 0.5 g/L quinine in EtOH after limited access drinking (*p* > 0.05; paired *t*-test; n.s.).

**Figure 6 brainsci-12-01042-f006:**
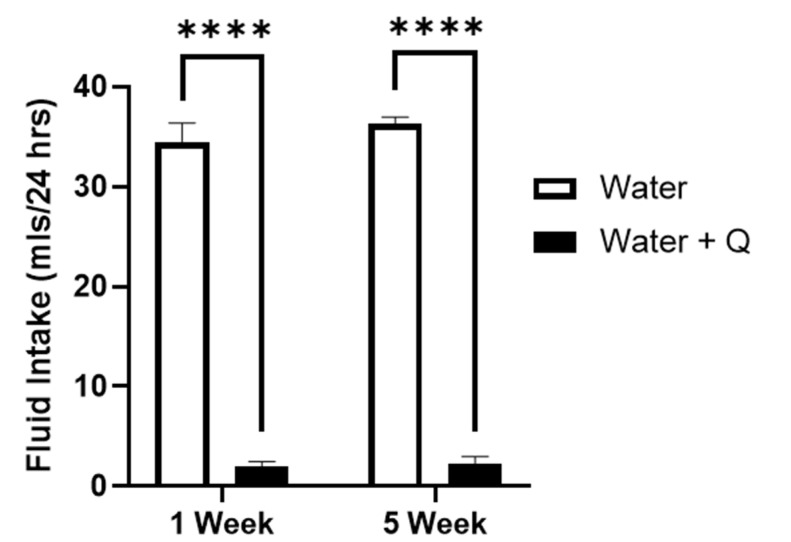
Quinine sensitivity in water. Male P-rats consumed EtOH under free-choice conditions for 1 week (left, *n* = 8) or 5 weeks (right, *n* = 7), then were tested under two-bottle choice conditions, with one bottle containing water, and the second bottle with 0.1 g/L quinine in water. Rats in both groups highly preferred water alone compared to the water adulterated with 0.1 g/L quinine (*t*-values > 16; **** *p* < 0.0001).

## Data Availability

At the time of publication, all data will be made available upon request.

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
