# Peer review of "Modeling Aversion Resistant Alcohol Intake in Indiana Alcohol-Preferring (P) Rats"

_brainsci, 2022, doi:10.3390/brainsci12081042_

Round 1
Reviewer 1 Report
The manuscript entitled: MODELING AVERSION RESISTANT ETHANOL INTAKE IN INDIANA P ALCOHOL-PREFERRING RATS by Katner et al. sought to describe the aversion-resistant drinking (ARD) behavior in P-bred rats from both genders. While the importance of the compulsive aspect of alcohol drinking despite negative consequences is of high importance in AUD as mentioned by the authors, the set of experiments presented in the ms. lack of many levels of study that deserve to approach in a better and more wide fashion, as explained below:
- Keywords: please use one or two-words instead of a phrase which better fit in a highlights section. In addition, the gender as a factor under study should be mentioned here or in the title.
- Material and methods:
Overall, no reference to the figures should be done in this section. Please, remove them to be referred in the results section.
2.1 please include the total number of animals used in the study. Although the number of animals assigned to each experiment is mentioned, it is not clear whether they are different subsets and if so, please explain the rationale to use different animals when the studies are sequential or progressively longer in terms of time of ethanol intake. In the same sense, were the animals sacrificed at the end of each experiment? and if that is the case, it should be desirable that any kind of neurochemical/biochemical/pharmacological approach would be included. In fact, this is the main pitfall of the study as the brain tissue of the animals under study is a precious sample to complement the behavioral data. This reviewer strongly suggests to complement the data with any of these end points.
In addition, no reference to the initial weigh of the animals is mentioned nor the weight gain across the experiment which should be considered provided the empty calories of ethanol. The day of the estrous cycle that the experiments with females begun must be taken into account because is an important factor in terms of reinforcement.
2.9 The statistical analysis performed in each experiment is not clear as stated in this section as no specific mention is written in each data result. More importantly, the F value, the degrees of liberty and the p value should be explicit in each comparison and fits better in the results section rather than in the legend which should only provide the p values obtained from the post hoc analysis. For instance, the legend in Fig 1 provides a value that is not clear in terms of the degree of freedom as two numbers are subscripted. At this respect, the correct statistical analysis for Figs 1 and 4-6 is a repeated-measures ANOVA with the t analysis not acceptable as a pos hoc test provided that 4 different groups are evaluated.
Results
All figures show a different rationale for the presentation style. Some have legends in form of boxes in the right side (Fig 6), some mentioned ethanol consumption alone as controls (Fig 1), in others the 15% ethanol mention is written (Fig 5) etc. This must be corrected. Some suggestions taking Fig 1 as an example is detailed below.
Fig 1. Increasing quinine concentrations were used (0-0.4 g/l). As the 0.5g/l dose was used in some of the following experiments it should have been desirable to have include that. However, provided that 0.4 g/l dramatically reduced ethanol intake, the same effect is anticipated for the higher dose. In addition, the Control bar represents the mean intake averaged from 4 days so it should be differentiated somehow from the other bars (i.e. as an insert or separated by dotted lines in parallel to the y axis).
Conclusion
The sentence: “Therefore, modeling ARD in P-rats may provide insight to compulsive drinking and lead to new prevention strategies and treatments for AUDs” stress the need of a biochemical or pharmacological approach that complements the present study.---
Author Response
The manuscript entitled: MODELING AVERSION RESISTANT ETHANOL INTAKE IN INDIANA P ALCOHOL-PREFERRING RATS by Katner et al. sought to describe the aversion-resistant drinking (ARD) behavior in P-bred rats from both genders. While the importance of the compulsive aspect of alcohol drinking despite negative consequences is of high importance in AUD as mentioned by the authors, the set of experiments presented in the ms. lack of many levels of study that deserve to approach in a better and more wide fashion, as explained below:
Response: We agree that additional study is needed and have indicated this in the manuscript, and in fact we are currently conducting follow-up studies to investigate many of these questions. However, the finding of inherent and rapid development of ARD in this line of rats selected for alcohol preference presented in this manuscript is novel and of great importance to the field and we consider it an impactful and publishable unit.
- Keywords: please use one or two-words instead of a phrase which better fit in a highlights section.
Response: We have done this.
In addition, the gender as a factor under study should be mentioned here or in the title.
Response: We have added this here as indicated.
Material and methods:
Overall, no reference to the figures should be done in this section. Please, remove them to be referred in the results section.
Response: All references to Figures have been removed from this section.
2.1 please include the total number of animals used in the study.
Response: We have provided the total numbers of animals as requested. Although the number of animals assigned to each experiment is mentioned, it is not clear whether they are different subsets and if so, please explain the rationale to use different animals when the studies are sequential or progressively longer in terms of time of ethanol intake.
Response: Only the male rats in Figure 5 were sequentially tested after finding the development of resistance to 0.5 g/L after 5 weeks of drinking and the need to determine if exposure to scheduled access drinking affected aversion resistance. We have now this information in this section. All the other experiments were conducted in separate cohorts to answer specific questions about the development of aversion resistance.
Response: In the same sense, were the animals sacrificed at the end of each experiment? and if that is the case, it should be desirable that any kind of neurochemical/biochemical/pharmacological approach would be included. In fact, this is the main pitfall of the study as the brain tissue of the animals under study is a precious sample to complement the behavioral data. This reviewer strongly suggests to complement the data with any of these end points.
Response: Unfortunately, we do not have brain tissue from this group of animals. Since this was the first study to fully characterize aversion resistant drinking (ARD) in the alcohol-preferring (P) rat in a 24 hr free-choice paradigm, identification of groups for tissue collection were not anticipated using an appropriate hypothesis-driven approach. However, neurochemical analysis of the effects are currently underway in follow-up studies in separate groups of animals using the alcohol exposure paradigms now identified in this study.
In addition, no reference to the initial weigh of the animals is mentioned nor the weight gain across the experiment which should be considered provided the empty calories of ethanol.
Response: We have now provided the approximate mean weights of the animals at the beginning of the experiments. The majority of experiments utilized a within-subject design or we did not have the ability to conduct an “apples to apples” comparison of body weight. However, in the first experiment, a simultaneous control group was run. Examining body weight gain from the beginning to end of the experiment, no significant between-groups differences in body weights were found. We have now included these data in the results section.
The day of the estrous cycle that the experiments with females begun must be taken into account because is an important factor in terms of reinforcement.
Response: We did not directly monitor the estrous cycle in this study and agree with the reviewer that it is a limitation of this study and have now indicated that additional work is needed in this area, as well as impacts on development, in the Discussion.
2.9 The statistical analysis performed in each experiment is not clear as stated in this section as no specific mention is written in each data result. More importantly, the F value, the degrees of liberty and the p value should be explicit in each comparison and fits better in the results section rather than in the legend which should only provide the p values obtained from the post hoc analysis. For instance, the legend in Fig 1 provides a value that is not clear in terms of the degree of freedom as two numbers are subscripted.
Response: We have engaged Dr. Leah Wetherill, as statistician at IUSM and have now included her in the authors list. We have now removed the subscripts and listed DF as they appear in the output.
At this respect, the correct statistical analysis for Figs 1 and 4-6 is a repeated-measures ANOVA with the t analysis not acceptable as a pos hoc test provided that 4 different groups are evaluated.
Response: We have provided/expressed the noted statistical information as requested. We have also added additional information on the RMAnova. We also added the missing info on our post-hoc tests as indicated.
Results
All figures show a different rationale for the presentation style. Some have legends in form of boxes in the right side (Fig 6), some mentioned ethanol consumption alone as controls (Fig 1), in others the 15% ethanol mention is written (Fig 5) etc. This must be corrected. Some suggestions taking Fig 1 as an example is detailed below.
Fig 1. Increasing quinine concentrations were used (0-0.4 g/l). As the 0.5g/l dose was used in some of the following experiments it should have been desirable to have include that. However, provided that 0.4 g/l dramatically reduced ethanol intake, the same effect is anticipated for the higher dose. In addition, the Control bar represents the mean intake averaged from 4 days so it should be differentiated somehow from the other bars (i.e. as an insert or separated by dotted lines in parallel to the y axis).
Response: Although each experiment is distinct and thus, some have different controls [i.e. control rat intakes were not monitored each day in Fig. 1, thus an average was used for the same period for those animals and, as noted correctly, could not be directly compared to the single day intakes measured in the other group]. We have adjusted the figures to clarify where possible, provided more statistical details in the Results section, and added the dotted line for Fig 1 as suggested.
Conclusion
The sentence: “Therefore, modeling ARD in P-rats may provide insight to compulsive drinking and lead to new prevention strategies and treatments for AUDs” stress the need of a biochemical or pharmacological approach that complements the present study.
Response: We fully agree with this comment. As such, this manuscript will serve as the basis for studies of the mechanisms underpinning how selection for alcohol preference may affect ARD.
Reviewer 2 Report
Thank you for giving me this opportunity to review the manuscript.
The manuscript submitted for publication by Katner et al., titled: "Modeling aversion resistant ethanol intake in Indiana P alcohol-preferring rats" is an observational study aiming to assess the quinine-resistant drinking in Indiana P alcohol-preferring rats.
The topic is interesting. The method and sample size is accepted, by and large.
The manuscript is simple, clear, well written and pleasant to read.
The statistical analysis adequate to the experimental protocol.
The manuscript needs to be improved considering the comments that are listed below:
1. The title of the manuscript should be edited. I would replace "Ethanol" with "Alcohol" (even if repeated within the title), because ethanol is never repeated throughout the manuscript. Or I would replace "Ethanol Intake" with "Drinking".
2. Similarly, in the Abstract and in all manuscript I would change "Ethanol (EtOH)" with "Alcohol".
3. In the introduction (line 39), as well as in Discussion (line 309), the authors refer to the use of alcohol in women and the genetic and familial predisposition to compulsive alcohol use. I believe it is worth commenting on the clear and interesting evidence of the effects of prenatal alcohol exposure on embryonic development and related neurological deficits, as well as the complex interplay of fetal and parental genes that can predispose offspring to compulsive alcohol use (Mead, E. A., & Sarkar, D. K. (2014). Fetal alcohol spectrum disorders and their transmission through genetic and epigenetic mechanisms. Frontiers in genetics, 5, 154; Castelli V, Brancato A, Cavallaro A, Lavanco G, Cannizzaro C. Homer2 and Alcohol: A Mutual Interaction. Front Psychiatry. 2017 Nov 30; 8: 268; Brancato A, Plescia F, Lavanco G, Cavallaro A, Cannizzaro C. Continuous and Intermittent Alcohol Free-Choice from Pre-gestational Time to Lactation: Focus on Drinking Trajectories and Maternal Behavior. Front Behav Neurosci. 2016 Mar 3; 10: 31; Sarkar, D. K., Gangisetty, O., Wozniak, J. R., Eckerle, J. K., Georgieff, M. K., Foroud, T. M., Wetherill, L., Wertelecki, W., Chambers, C. D., Riley, E., Zyma k-Zakutnya, N., & Yevtushok, L. (2019). Persistent Changes in Stress-Regulatory Genes in Pregnant Women or Children Exposed Prenatally to Alcohol. Alcoholism, clinical and experimental research, 43 (9), 1887–1897).
4. The authors do not explain why they focused on effects in adulthood (PND63) and not in adolescence. It would be interesting to observe ARD in adolescence, being a highly sensitive period of life.
5. I believe it is necessary to modify or even delete the sentence in the discussion (line 277) "Modeling ARD in P-rats may provide insight to compulsive drinking and lead to new prevention strategies and 278 treatments for AUDs." It is repeated several times, in the abstract and in the conclusions.
6. It might be a plus to the manuscript to add a small paragraph on the future perspectives of using this Indian animal model, which appears to be more resistant to high concentrations of quinine than other experimental animal models.
Author Response
- The title of the manuscript should be edited. I would replace "Ethanol" with "Alcohol" (even if repeated within the title), because ethanol is never repeated throughout the manuscript. Or I would replace "Ethanol Intake" with "Drinking".
Response: We have made the requested change in the title.
- Similarly, in the Abstract and in all manuscript I would change "Ethanol (EtOH)" with "Alcohol".
Response: We have now defined and abbreviated alcohol by “EtOH”..
- In the introduction (line 39), as well as in Discussion (line 309), the authors refer to the use of alcohol in women and the genetic and familial predisposition to compulsive alcohol use. I believe it is worth commenting on the clear and interesting evidence of the effects of prenatal alcohol exposure on embryonic development and related neurological deficits, as well as the complex interplay of fetal and parental genes that can predispose offspring to compulsive alcohol use (Mead, E. A., & Sarkar, D. K. (2014). Fetal alcohol spectrum disorders and their transmission through genetic and epigenetic mechanisms. Frontiers in genetics, 5, 154; Castelli V, Brancato A, Cavallaro A, Lavanco G, Cannizzaro C. Homer2 and Alcohol: A Mutual Interaction. Front Psychiatry. 2017 Nov 30; 8: 268; Brancato A, Plescia F, Lavanco G, Cavallaro A, Cannizzaro C. Continuous and Intermittent Alcohol Free-Choice from Pre-gestational Time to Lactation: Focus on Drinking Trajectories and Maternal Behavior. Front Behav Neurosci. 2016 Mar 3; 10: 31; Sarkar, D. K., Gangisetty, O., Wozniak, J. R., Eckerle, J. K., Georgieff, M. K., Foroud, T. M., Wetherill, L., Wertelecki, W., Chambers, C. D., Riley, E., Zyma k-Zakutnya, N., & Yevtushok, L. (2019). Persistent Changes in Stress-Regulatory Genes in Pregnant Women or Children Exposed Prenatally to Alcohol. Alcoholism, clinical and experimental research, 43 (9), 1887–1897).
Response: We have added this topic/refs to the Discussion.
- The authors do not explain why they focused on effects in adulthood (PND63) and not in adolescence. It would be interesting to observe ARD in adolescence, being a highly sensitive period of life.
Response: We began the study in adulthood to compare with data from other strains (most from adults). However, we plan to investigate adolescents in the future for the reasons indicated.
- I believe it is necessary to modify or even delete the sentence in the discussion (line 277) "Modeling ARD in P-rats may provide insight to compulsive drinking and lead to new prevention strategies and 278 treatments for AUDs." It is repeated several times, in the abstract and in the conclusions.
Response: We have removed this line from the first paragraph of the Discussion.
- It might be a plus to the manuscript to add a small paragraph on the future perspectives of using this Indian animal model, which appears to be more resistant to high concentrations of quinine than other experimental animal models.
Response: We have added a statement to end the article to emphasize this point.